# The global polarisation of remote work

**Fabian Braesemann**[1,2,3] *, **Fabian Stephany**[1,2,3], **Ole Teutloff**[3,4], **Otto Kässi**[1,5], **Mark Graham**[1], **Vili Lehdonvirta**[1]

**1** Oxford Internet Institute, University of Oxford, Oxford, United Kingdom, **2** Humboldt Institute for Internet and Society, Berlin, Germany, **3** DWG Datenwissenschaftliche Gesellschaft Berlin, Berlin, Germany, **4** Copenhagen Center for Social Data Science, University of Copenhagen, Copenhagen, Denmark, **5** Etla Economic Research Helsinki, Helsinki, Finland

\* fabian.braesemann@oii.ox.ac.uk

**Data Availability Statement:** The data to replicate the findings presented in the study is publicly available on GitHub: https://github.com/Braesemann/remotework.

## Abstract

The Covid-19 pandemic has led to the rise of digitally enabled remote work with consequences for the global division of labour. Remote work could connect labour markets, but it might also increase spatial polarisation. However, our understanding of the geographies of remote work is limited. Specifically, in how far could remote work connect employers and workers in different countries? Does it bring jobs to rural areas because of lower living costs, or does it concentrate in large cities? And how do skill requirements affect competition for employment and wages? We use data from a fully remote labour market—an online labour platform—to show that remote platform work is polarised along three dimensions. First, countries are globally divided: North American, European, and South Asian remote platform workers attract most jobs, while many Global South countries participate only marginally. Secondly, remote jobs are pulled to large cities; rural areas fall behind. Thirdly, remote work is polarised along the skill axis: workers with in-demand skills attract profitable jobs, while others face intense competition and obtain low wages. The findings suggest that agglomerative forces linked to the unequal spatial distribution of skills, human capital, and opportunities shape the global geography of remote work. These forces pull remote work to places with institutions that foster specialisation and complex economic activities, i. e. metropolitan areas focused on information and communication technologies. Locations without access to these enabling institutions—in many cases, rural areas—fall behind. To make remote work an effective tool for economic and rural development, it would need to be complemented by local skill-building, infrastructure investment, and labour market programmes.

## Introduction

The Covid-19 pandemic has made remote work the 'new normal' form of organisation for many white collar employees. Pre-covid, only the most flexible employers allowed employees to work at a distance. The coordination costs of managing remote teams were considered too high [1]. In forcing office employees to work from home, the pandemic has vastly accelerated the adoption of digital technologies [2] and organisational adjustments that allow business processes to operate productively at a distance [3].

**Funding:** FB received funding from the John Fell Oxford University Press Research Fund, grant number 0008391 (https://researchsupport.admin.ox.ac.uk/funding/internal/jff) Mark Graham received funding from the European Research Council H2020, grant number 335716 (https://ec.europa.eu/programmes/horizon2020/en/h2020-section/european-research-council) Vili Lehdonvirta received funding from the European Research Council H2020, grant number 639652 (https://ec.europa.eu/programmes/horizon2020/en/h2020-section/european-research-council) Fabian Stephany received funding from the Economic and Social Research Council, grant number G2781-15 (https://www.ukri.org/councils/esrc/) The funders had no role in study design, data collection and analysis, decision to publish, or preparation of the manuscript.

**Competing interests:** The authors have declared that no competing interests exist.

Organisations can realise substantial cost savings and tap into global talent pools if they adopt remote working practices and start to outsource business processes to the remote workforce [4]. But what are the consequences of remote working for the global division of labour? Here, we investigate the global geographies of a fully remote labour market—a so-called online labour platform—that provides the digital infrastructure to hire remote workers on demand. These online platforms have been established over the past 10 to 15 years and they allow even small companies or individual employers to outsource knowledge work to individual freelancers [5]. Due to the digital organisation of the hiring and work process on the platform, online labour markets can be considered as a prototype of a fully remote labour market. Having started as niche marketplaces for IT freelancers, these platforms now cover the whole spectrum of knowledge work, from data entry to management consulting, with millions of platform workers involved globally [6–8], and rising adoption during the Covid-19 pandemic [9]. The platform labour market has in fact seen substantial growth in recent years. According to Kässi et al. (2021) the number of globally registered online workers was 163 million in 2021 [6]. With the decade-long shift towards remote work [10] that has only been accelerated by the pandemic, more of the overall labour market could begin to resemble the online labour market soon.

Remote work organised via online platforms could bring jobs to workers from all over the globe [11–13]. In doing so, emote work could help to mitigate the global imbalance between the increasing supply of, and competition between, highly educated graduates in Global South and Global North countries—a phenomenon which has been described as the 'Global Auction' [14]—and the high global demand for talent. In bringing jobs and income to people in countries across the world and, in particular, in rural areas, remote work could help to foster more resilient, sustainable local communities [15, 16] and offer alternatives to the physical migration to places with more jobs and higher wage levels [17], if platform work can provide sustainable sources of income. However, several studies have reported that remote platform work is shaped by geographical frictions and biases that restrict participation, for example, time-zone differences, language-based communication difficulties, domestic and ethnic connections, or information asymmetries [18–21]. Similar to other complex economic activities such as research, innovation and industry [22] remote platform work might cluster in large cities. In the global remote labour market, modularisation of tasks and competitive dynamics could cause uneven geographical participation rates [23], bad working conditions [24–26], and precariousness for workers [27–29]; a process that has been subsumed under the term 'Digital Taylorism' [30]. Overall, the individual contributions provide coherent, but fragmented perspectives on the phenomenon of remote platform work. The body of previous contributions in sum, however, does not provide a comprehensive and stringent picture with respect to the global geography of remote platform work and its' impact on the relationship between urban and rural areas. One reason for this is a lack of sufficiently granular data.

Here, we examine the global geography of remote freelance work mediated by online labour platforms empirically based on a data set covering 1.8 million remote jobs from 2013 to 2020 to show that remote platform work mirrors the geographical and skill-based polarisation of labour markets at large [31]. The data comes from one platform, which is among the largest in terms of global market shares [8, 32]. In the discussion section, we consider what implications our findings may have for remote platform work more generally and other types of remote work, including regular employment performed remotely over the Internet.

In mapping the platform jobs to sub-national geographies in 139 countries and an established occupation taxonomy, we reveal that the remote labour market is polarised along three dimensions: globally between countries, between urban and rural areas within countries, and, overall, between occupations and skill-sets. We relate the spatial and occupational variation to

differences in the global distribution of infrastructure, economic opportunities, and human capital. The data suggest that agglomerative forces shape the global geography of remote work. These forces pull the most profitable remote jobs to metropolitan areas and locations with existing competitive advantages in information and communication technologies. At the same time, rural areas and disadvantaged regions, particularly in Global South countries, are not able to attract many remote jobs. Urban specialists can realise a premium for scarce skills [33], while less specialised remote platform workers compete for low-wage jobs.

Based on these findings we argue that the global distribution of skills and enabling institutions are the focal point of remote work. In this perspective, Digital Taylorism—the standardisation and modularisation of complex production processes of the knowledge economy broken down into simple and codified tasks together with improved monitoring capabilities [27]—is the very process that makes remote work and global digital value chains possible [14]. Enabling cost savings and access to talent pools simultaneously, Digital Taylorism is considered as the driver of the specialisation and global integration of the digital workforce via remote work. This process affects incomes and opportunities of knowledge work globally.

According to this reading of the empirical findings, we provide the following interpretation of the global geography of platform work. Skill-biased technological change [34–36] allows people with advanced digital skills (e. g. Data Scientists) to realise a premium from increased demand, while offshoring, computerisation, and global competition for jobs that require less specialised knowledge (e. g. Data Entry) drive wages downwards [37, 38]. The result is a polarised global market for knowledge work [39] with its' geography stratified along the lines of the unequal distribution of human capital. The antagonism between urban and rural areas—described by Paul Collier in his book *The Future of Capitalism* as a deep dividing line between the 'booming metropolis' and the 'broken provincial city' [40]—plays out fully in the remote labour market. This is because the institutions that enable a successful participation—access to knowledge building, training and professional networks—concentrate in urban environments. Rural regions are less able to offer specialised work opportunities and urban lifestyle [41, 42]. In contrast, in metropolitan areas, a highly specialised local economy creates an abundance of opportunities to maintain a tech-savvy 'creative class' [43, 44]. The most profitable remote jobs require specialised IT-skills and go therefore to metropolitan areas. The polarising forces that pull remote jobs to centres of economic activity with existing competitive advantages and digital infrastructures work almost unrestricted in the platform economy. This is because there are only limited frictions of geographical boundaries, labour market regulations or formal entry barriers, which increase the role of information asymmetries, uncertainties, trust cues, and reputation systems in online labour markets [45–47].

In summary, our main argument is that the unequal global distribution of remote work is the result of the unbalanced distribution of skills, human capital, and opportunities across the globe [48, 49]. This uneven distribution of economic conditions and competitive advantages transcends to the platform economy and drives the geographical polarisation of the remote labour market. Remote workers with access to hard-to-copy skills can realise a substantial premium, while those who lack marketable, digital skills participate in a global rat race for remote jobs. Digitally enabled remote work will likely not disperse knowledge work more evenly across space but rather reinforce prevalent agglomerative dynamics.

## Literature review

In the following, we review the existing literature on (a) the relationship between digital technologies and changing economic geographies, and (b) previous empirical approaches to measure platform work. The overview of the literature presented here is complemented by a more

extensive literature review in the Supplementary Information (S3 Appendix: S3 Section, in particular S1 to S5 Tables in S1 File). In the appendix, we also provide more context on definitions and the conceptualisation of the term 'remote platform work' (S2 Appendix in S1 File). Here, we focus on a brief discussion of common findings, data, methods, and limitations of previous work from which we derive the contribution of our study.

## The economic geography of outsourcing, offshoring, and digital technologies

The rise and role of ICT technologies for (global) economic geographies is a subject of extensive scholarly work and debate. Through outsourcing, offshoring and task automation, ICT technologies profoundly affect the location of economic activities and the global dynamics of interaction between economic centres. The internet has reduced frictions of economic interactions: low communication, transportation, and search costs, for example, enable more densely connected global value chains [50, 51].

Digital technologies facilitate the trade of commodities and in particular the trade of services at a distance. The offshoring of services allows for more complex global production networks [52] which further increase the international division of labour [53]. Several studies have attempted to capture the share and nature of jobs that could be offshored thanks to the increasing technological capabilities [34, 37, 38, 54]. While digital labour platforms are a relatively recent phenomenon, the offshoring of service activities has a long history. For example, in 1983, American Airlines established its first international back office in Barbados [55]. Paperwork was flown from the U. S., handled and digitised in Barbados and sent back electronically via satellite to save 50 per cent labour costs. In 2017, Feakins described how offshoring of services enabled complex global economic relationships, such as the Off-offshoring of service tasks from Russia to Ukraine [56].

ICTs are also reshaping the distribution of economic activities within countries [50]. For instance, Malecki (2002) observes that the geography of Internet backbone networks reinforces urban-rural differences [57]. Despite considerable controversy, the evidence overall suggests that the productivity impacts of ICTs are positive, but unevenly distributed [58, 59]. As a result, urban centres are likely to benefit disproportionately from digital technologies, as they tend tend to host specialised institutions and a more fine-grained division of labour than rural areas [22, 23, 60]. These are considered as enablers of complementary skills necessary to capitalise on the opportunities of digital technologies [61, 62]. The same holds for the integration of Global South countries into digital value chains. Overall, Hjort and Poulsen (2019) show that fast Internet access correlates positively with employment rates in African countries, due to the technology's impact on firm entry, productivity, and exports [15]. However, for example, Foster et al. (2018) find that East African firms involved in global value chains can only fully benefit from improved internet connectivity if they have access to complementary capacities and competitive advantages.

In summary, the internet has profoundly affected the geography of economic activities within and between countries across the globe over the past two to three decades. Overall, Information and Communication Technologies in general and the internet in particular have supported more fine-grained production chains and the offshoring of service activities. Online labour platforms continue this trend: they could represent one avenue for the further integration of remote workers into global digital value chains through outsourcing and offshoring. Studies from the platform economy literature have researched the geography of this novel form of online mediated work. In the following, we discuss the main findings and limitations

of these studies with respect to the geography of remote platform work and the role of skills, and we derive our study's contribution from there.

## The economic geography of platform work

Empirical studies on the geography of platform work have mainly focused on (a) assessments of the size of the platform labour market, (b) analyses of trade between countries, and (c) the role of signalling mechanisms and skills.

The first overarching finding is that platform labour markets are growing in importance and size [5, 13, 63] with estimated yearly growth rates of around 20% [6, 8].

Secondly, studies have investigated international transactions in the platform labour market: these are dominated by north-south interactions with employers from industrialised countries and workers from less developed countries [12, 13, 21, 63, 64]. Despite its inclusive global digital infrastructure, several barriers to trade and sources of worker discrimination persist such as geographical distance, language, time zone as well as cultural and ethical differences [13, 19–21, 64].

However, most of these studies use relatively small or old datasets, are limited to a small set of countries or investigations on the country-level. For example, Horten et al. (2017) [13] focus on the bilateral trade between countries in the online labour market with an economic gravity model. Their investigation aims on statistically explaining the large share of US-Indian trade on the online platform, and it stays largely on a descriptive level. Similarly, Agrawal et al. (2015) [63] provide descriptive statistics about the development of wages in occupational groups and selected countries over time, but do not explain the wage differences quantitatively. Lehdonvirta et al. (2019) [5] theorise online labour platforms through transaction cost economics and test their hypotheses empirically with a data set of 10,000 projects in two countries. Borchert et al. (2018) [65] and Braesemann et al. (2022) [23] analyse the relationship between online labour participation and local economic conditions on a sub-national level, but their investigations are limited to only one country.

These examples highlight a limitation of previous empirical contributions on the geography of platform work: many studies used relatively small cross-section data sets and they treated countries as the smallest unit of geographical analysis. This impeded a thorough identification of the economic determinants that shape the geography of online labour markets between countries and sub-national regions on a global level.

A third stream of research has focused on the role of skills in remote platform work. These studies find that while skills are considered as an important predictor of wages, workers seem to have limited opportunities to learn and grow on online platforms [12, 27, 28, 65]. Skill certificates can increase worker earnings [32]. However, there is contradicting evidence about the role of reputation systems in building trust, ranging from having an inclusive effect benefiting workers from developing countries disproportionately [21] to reputation leading to increasing inequality ("super star effect") [66]. Details of the platform design seems to play an important role.

In the studies on skills and signalling mechanisms, the operationalisation of skills represents a common challenge: many of the reviewed studies present simplistic operationalisations of skills or do not explicitly measure them at all. For example, Kässi and Lehdonvirta (2022) [32] focus on the wage effects of online workers obtaining skill certificates rather than on skill-based or occupational differences. Lehdonvirta et al. (2019) [5] compare only two occupations (writing vs. graphic design), similarly to Beerepoot and Lambregts (2015) [12] who operationalise differences in skill levels with only two occupations (web development = high skill vs. administrative support = low skill). Other studies focus more on reputation

mechanisms than on skills or occupations as individual-level wage determinants [19, 66–68]. The network approach of Anderson (2017) [33] represents an exception. Anderson proposes a network-based method for measuring worker skills and finds that workers benefit from skill diversity.

In summary, while previous studies have investigated geographical and skill-related aspects of platform work, there is still a gap in the literature regarding the global distribution of online labour, and in particular its' relation to the unequal global distribution of skills, human capital, and place-bound institutions. We extend the current literature on global geography of remote platform work by a quantitative analysis based on three relevant methodological advances: more and comprehensive data, geographic granularity, and matching of skills.

First, we collect one of the most extensive datasets on the subject, including almost 2 million projects spanning eight years from 2013—2020 and a fine-grained geographical granularity. Secondly, we take a global viewpoint including 139 countries while extending the analysis to the sub-national level. In particular, we map the geocoded online platform data to regional statistical data from the World Bank, the OECD and the Global Data Lab on the country- and sub-national level, which makes the global persistence of urban-rural differences visible for the first time. This step also allows for applying regression modelling to explain the geographical distribution in relation to local economic conditions and infrastructure. Thirdly, we propose a sound methodology to operationalise skills by matching remote jobs to a well-established occupational taxonomy. These three methodological advances allow us to pose the following research questions:

**RQ 1** How is remote platform work globally distributed between and within countries? What is the relationship between urban and rural areas?

**RQ 2** Are the unequal global distributions of remote platform work activity and wage differentials in Global North and Global South countries determined by local institutions, infrastructure and economic conditions?

**RQ 3** In how far are outcomes of different job types in the remote platform labour market (wage levels, value of experience) related to differences in competitive intensity and skill requirements across occupations?

## Data and methods

All the methods of the data collection and analysis are described in great detail in the Supporting Information (S4 and S5 Appendices in S1 File). Here, we summarise the most critical steps.

### Data collection

In this study, we combine three data sources: (a) transaction records from a globally leading online labour platform, (b) regional covariates covering the demography, economy and infrastructure in OECD+ and Global South countries, and (c) occupation statistics from the U. S. Bureau of Labour Statistics (S4 Appendix in S1 File). The retrieval and assembly of online labour records proceeds in two steps: After having gathered information via the API about the projects of which we had IDs, we extracted the platform worker IDs from these projects. In total, we collected a data set of 1.8 million remote jobs from one global online labour platform covering eight years from 2013 to 2020.

## Geocoding

In a second step, we provided these IDs to the API to obtain the remaining information related to each transaction of these platform workers. This includes the hourly wage, the total price charged for the project, and the workers' country-city location (S4 Appendix: S4.1 Section in S1 File). Afterwards, we use a Geocoding API and provide it with a list of all unique country-city locations from both the employer and worker side of the platform transactions; a total of 66,085 locations (S4 Appendix: S4.2 Section in S1 File). Then, the geocoded online labour data is matched with national and sub-national statistics on demography, economy, and infrastructure. Here, three data sources are considered: World Bank for country-level statistics, OECD regional statistics for sub-national data in high and middle income countries from the Global North, and Global Data Lab for sub-national level data for low and middle income countries from the Global South (S4 Appendix: S4.3 Section in S1 File).

## Occupation mapping

Besides the geographical analysis of online labour data, we also investigate the job types of the online projects. For this purpose, we match the online job categories with official occupational statistics used by the U. S. Bureau of Labour Statistics (BLS). The BLS provides detailed information about each occupation's educational requirements, skills, and abilities. This data is available via the Occupational Information Network O*NET (S4 Appendix: S4.4 Section in S1 File). To match online work descriptions with official occupational taxonomies, we use the SOCcer (Standardized Occupation Coding for Computer-assisted Epidemiological Research) tool provided by the U. S. National Institutes of Health for an automatised coding of a sample of 345,000 online projects. We provide the online job category as job title, and the required skills and description of the online project as job description to the tool. Based on the occupational mapping, we derived two measures 1) capturing the skill or educational requirements of different occupations from BLS data and 2) the relevance of experience in obtaining online projects (S4 Appendix: S4.5 Section in S1 File). We present the distribution of skill requirements and use hierarchical cluster, applying a Euclidean distance measure and complete linkage as clustering method, to group skills and occupations. Furthermore, we relate occupation-level variables to the importance of experience in obtaining online projects. This is what we call the 'experience gradient'. The idea is that experience and reputation are known to drive outcomes in the platform economy, as they signal trustworthiness of sellers.

## Regression analysis

Lastly, the regression analysis of the geographical distribution of online labour projects and wages relies on six regression models with multilevel effects, where we regress a broad set of regional characteristics on regional- and country-level wages and project count whilst ensuring accurate feature selection and out-of-sample prediction accuracy (S5 Appendix in S1 File), including various robustness checks to ensure that our findings are consistent across time and space (S6 Appendix in S1 File).

## Results

### Polarisation across space

Considering Research Question 1 on the global distribution of remote platform work, the data shows that the online labour market connects global demand for and supply of remote knowledge work (Fig 1). However, while it is theoretically open to users from all over the world, Fig 1A shows that demand and supply highly cluster in a limited number of places. Most demand

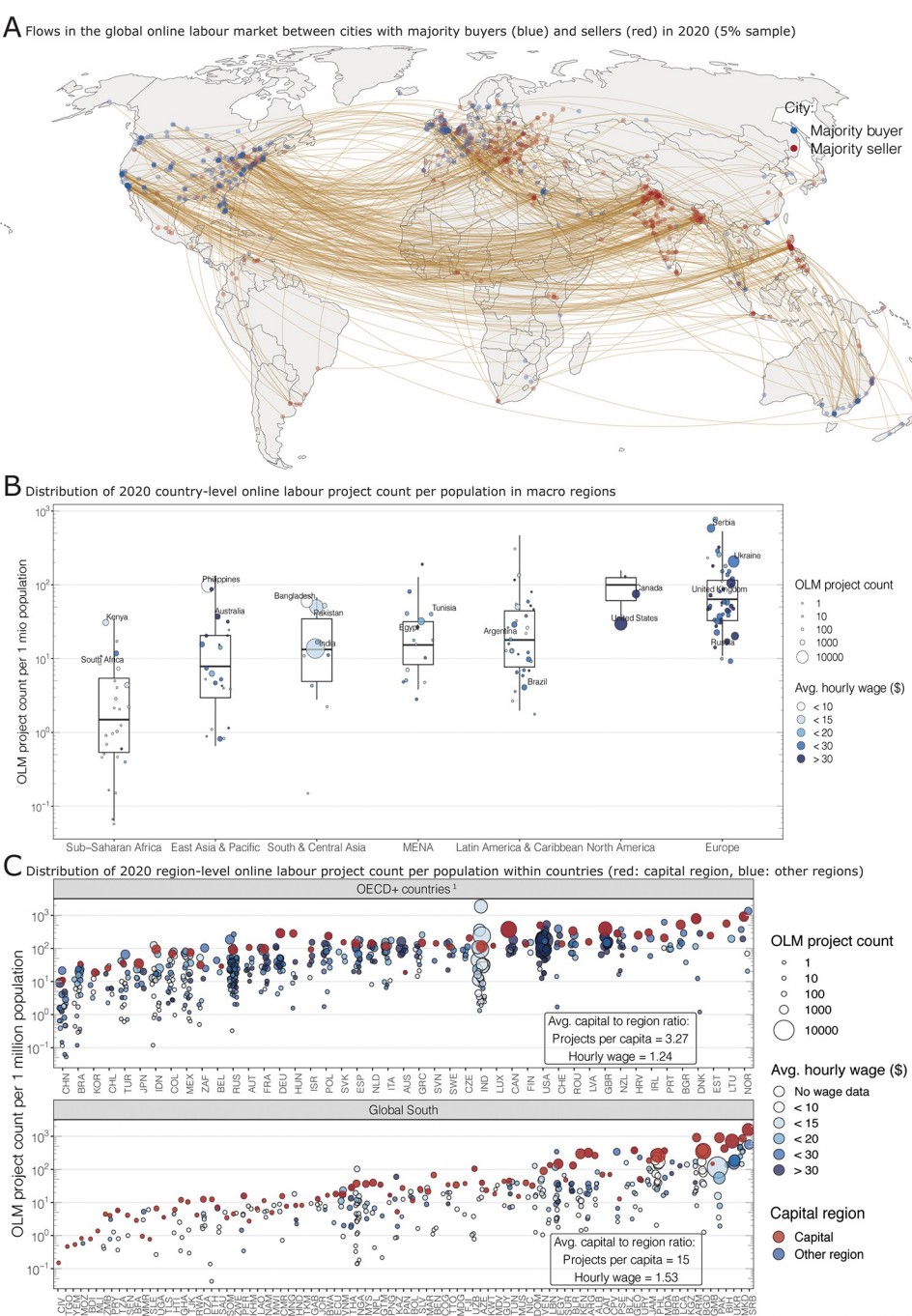

**Fig 1. (A)** Connections between majority buyer (blue) and seller cities (red) in the 2020 platform labour market (5% sample): hotspots of demand are North America, West Europe, and Australia; platform workers in Eastern Europe, South Asia and the Philippines conduct most remote jobs. **(B)** Distribution of 2020 online labour (OLM) project count per capita (y-axis) in countries (dots), grouped by global macro regions (x-axis): globally, platform activity varies by several orders of magnitude; Europe and North America show the highest levels of participation, and the highest average wages (dot colour); most countries in the Global South participate only marginally in the remote labour market with low wages and less than 10 projects per one million population, with the exception of the Philippines, Bangladesh, Pakistan, and India. **(C)** Online labour distribution within countries in OECD+ and Global South countries: participation varies vastly within countries with most capital regions (red) hosting the largest platform worker communities per country; the imbalance is particularly pronounced in the Global South where the capital region hosts, on average, 15-times as many projects per capita than other regions in the country.

comes from urban areas in North America, West Europe, and Australia; most remote platform workers are located in East Europe, South Asia, and the Philippines. Dense flows of capital and labour connect these regions, while many Global South countries only marginally participate in the remote labour market. The overall polarisation remains largely persistent over time (S6 Appendix: S6.5 Section in S1 File).

The global differences in participation become more pronounced when considering the number of projects per capita (Fig 1B). Online labour project count per population varies by several orders of magnitude within and between macro-regions (position of the dots on the y-axis): while almost all countries in Europe and North America hosted at least ten projects per one million population in 2020, only half of the countries in South & Central Asia, one-third in East Asia & Pacific, and 15% in Sub-Saharan Africa did so. In absolute numbers (size of the dots), more than 50% of all online labour projects were conducted by platform workers from just five countries (India, Pakistan, Philippines, United States, Bangladesh). Hourly wages (colour of the dots) also vary substantially: while platform workers in the United States, United Kingdom, and Russia charged more than 30 USD per hour on average, remote workers in Bangladesh and the Philippines earned just a fifth, or 6 USD an hour. There are also exceptions: for example, Kenya and South Africa host relatively active platform worker communities. Kenya's participation per population is comparable to that of the United States. Averages wagers of South Africa's platform workers are comparable to those in Europe.

Inequalities between countries online labour activity have been reported in the past [13, 63]. However, here we also reveal the sub-national concentration of remote platform work globally (For more details, see S6 Appendix: S6.3 Section in S1 File). Participation rates vary by two to four orders of magnitude in many countries (Fig 1C), and the distribution within and between countries is highly concentrated, both in OECD + BRIICS (OECD countries and Brazil, Russia, India, Indonesia, China, and South Africa, abbreviated as OECD+). and Global South countries (S6 Appendix: S17 Fig in S1 File). In many countries, the capital region attracts most platform jobs in absolute and per capita terms (Fig 1C), particularly in the Global South. In the OECD+, the capital regions attract more than three times as many platform jobs per capita than other regions in the same county on average. In the Global South, capital regions obtain more than 15 times as many projects per capita as other regions in the same country. On a global scale, platform work is a metropolitan phenomenon.

Additionally, hourly wages vary substantially between metropolitan and other regions. On average, platform workers in OECD+ capital regions earned 24% more per hour than their counterparts in other regions. The wage spread was almost twice as high in Global South countries: platform workers in capital regions earned 53% more than those in other regions.

Summarising the findings related to Research Question 1, the data suggest two dimensions of geographical polarisation in the global remote labour market. First, we find pronounced inequalities between countries worldwide along a North-South dimension. Most online labour demand comes from high-income countries. Supply comes mainly from platform workers in traditional outsourcing destinations in South Asia, the Philippines, and by platform workers from Europe and North America. Secondly, we identify persistent inequalities within countries. This points towards urban-rural differences as the second main polarisation dimension in the remote labour market. Platform workers in large cities and places with enabling institutions seem to be able to secure more and better paid jobs than their counterparts in rural regions.

Turning to Research Question 2, we find that regional factors explain the spatial polarisation described in Fig 1. Fig 2 shows six regression models relating the number of projects and hourly wages per country (models 1 and 4), per OECD+ region (models 2 and 5), and per Global South region (models 3 and 6) to country-level and sub-national covariates. Data

**A** Regression models relating online labour project count (1-3) and avg. hourly wage (4-6) to regional covariates 2013 to 2020

| Dependent variable: | Yearly online labour project count[a] | | | Online labour avg. wage per hour[a,b] | | |
|---|---|---|---|---|---|---|
| Level: | Countries | Sub-national regions | | Countries | Sub-national regions | |
| Geography: | Global | OECD+[c] | Global South[d] | Global | OECD+[c] | Global South[d] |
| Model: | (1) | (2) | (3) | (4) | (5) | (6) |
| **Population**<br>Population, total (log scale) | **0.96***<br>(0.02) | **0.85***<br>(0.09) | **0.54***<br>(0.04) | **0.04***<br>(0.01) | 0.05<br>(0.04) | **0.06***<br>(0.02) |
| **Education**<br>Model (1), (4): share of pop. with secondary education<br>Model (2), (5): share of pop. with tertiary education<br>Model (3), (6): avg. years of education | **0.06***<br>(0.003) | 0.004<br>(0.01) | **0.07***<br>(0.03) | **0.01***<br>(0.001) | **0.01***<br>(0.002) | **0.06***<br>(0.02) |
| **Income per capita**<br>Model (1), (4): GDP per capita (in 1,000 $)<br>Model (2), (5): GDP per capita (2015 PPP $, log scale)<br>Model (3), (6): Gross National Income p. c. (2011 PPP $) | **−0.01***<br>(0.003) | **−0.33***<br>(0.10) | **0.06***<br>(0.02) | **0.002***<br>(0.001) | **−0.12***<br>(0.05) | **0.03***<br>(0.01) |
| **Internet connectivity**<br>Model (1), (4): fixed broadband subscriptions per 100 people<br>Model (2), (5): share of HHs with internet broadband access<br>Model (3), (6): share of HHs with internet access | **0.04***<br>(0.01) | **0.03***<br>(0.004) | **0.02***<br>(0.004) | **0.01***<br>(0.002) | **0.005***<br>(0.001) | **0.003****<br>(0.001) |
| **IT specialisation of the economy**<br>Model (1), (4): ICT share of all service exports (log scale)<br>Model (2), (5): Gross value added in ICT (2015 PPP $, log) | **0.22***<br>(0.04) | **0.56***<br>(0.05) | | **−0.02***<br>(0.01) | **0.09***<br>(0.02) | |
| **English language**<br>Indicator: English is official language | **0.68***<br>(0.11) | | | **−0.07****<br>(0.03) | | |
| **Price level**<br>PPP conversion factor (per 1,000 int. $) | **−0.32***<br>(0.08) | | | −0.04<br>(0.02) | | |
| **Capital region**<br>Indicator: region holds country capital | | 0.13<br>(0.10) | **1.75***<br>(0.10) | | **−0.08****<br>(0.03) | **−0.12****<br>(0.05) |
| Constant | | | | | **3.21***<br>(0.31) | **2.49***<br>(0.12) |
| n (regional units) | 139 | 292 | 305 | 112 | 253 | 56 |
| Observations | 1,136 | 2,384 | 2,122 | 763 | 1,536 | 255 |
| Fixed / Random Effects | Yearly FE | Country-year Fixed Effects | | Yearly FE | Country-year Random Effects | |
| R² | 0.72 | 0.70 | 0.43 | 0.44 | 0.79 | 0.69 |
| Adjusted R² | 0.71 | 0.67 | 0.36 | 0.43 | 0.79 | 0.68 |

Note:
*p<0.1; **p<0.05; ***p<0.01
[a] Inverse hyperbolic sine (ihs) transformed: $y = log(x + \sqrt{x^2 + 1})$.
[b] To ensure convergence of the mean wage, countries or regions with a project count of less than 25 have been excluded from the regression models (4)–(6).
[c] Regions in OECD countries and Brazil, Russia, India, Indonesia, China, and South Africa.
[d] Regions in Global South countries available in the *Global Data Lab* database.

**B** Spread of online labour project count (left panel) and average hourly wage (right panel) per country in 2020, compared to the spread of the residuals of the optimised models (1) and (4)

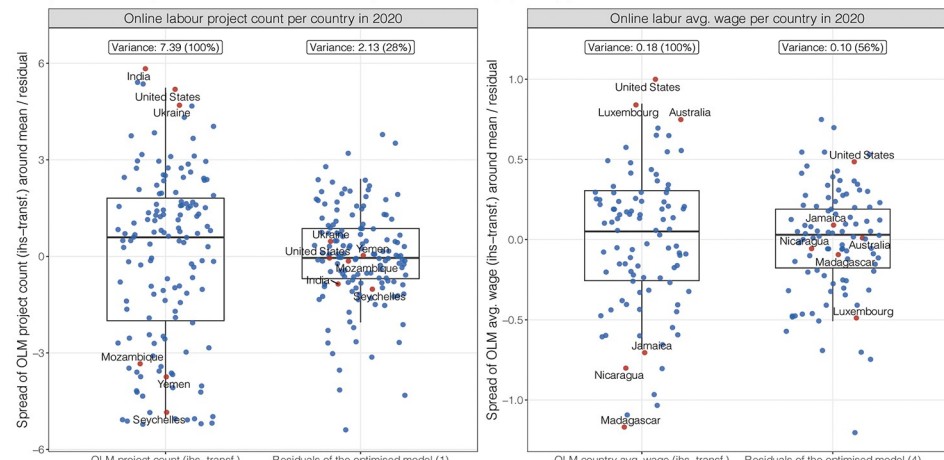

**Fig 2.** (**A**) Regressions between online labour project count (models 1—3), avg. hourly wage (models 4—6) and regional covariates (2013—2020 data, in total 1.76 million projects): population, education, internet connectivity, and the IT specialisation of the economy are positively associated with project count and hourly wages; globally, countries with English language and low price levels are more active in the remote labour market. (**B**) Spread of online labour (OLM) project count (left panel) and avg. hourly wage (right panel) per country vs. residuals of the regression models (1) and (4): the parsimonious models explain large shares of the global variation; for example, the countries at both ends of the project and wage spectrum (highlighted in red) show substantially reduced residuals after controlling for regional covariates. Overall, the regression models explain between 42% and 79% of the variation between countries or regions.

sources are World Bank data (countries), the OECD regional database (sub-national regions OECD+), and the Global Data Lab [69] (sub-national regions Global South). The data covers the years 2013—2020 (for details on the data set and pre-processing, see S4 Appendix: S4.1–S4. sections and S5 Appendix: S5.1 Section in S1 File). In summary, the regression models contain data of 139 countries an 597 sub-national regions over eight years.

The dependent variable in the models (1) to (3) is inverse hyperbolic sine (ihs) transformed ($y = log(x + \sqrt{x^2 + 1})$, see [70]) to reduce the skewness of the distribution, which ranges across several orders of magnitude. Hourly wages also vary substantially. Therefore, we also applied the ihs-transformation to the dependent variable in the models (4) to (6). As our models deal with different hierarchical levels (regions nested in countries or years), multi-level effects need to be considered. We test and apply random and fixed effects to account for the variability of outcomes within and across countries or years. (see S5 Appendix: S5:3 Section in S1 File).

To model the relation between the platform data and regional characteristics, we included regional statistics commonly used in studies on the platform economy (see S3 Appendix, S4 Appendix: S4.3 Section, and S5 Appendix: S5:2 Section in S1 File).

The regression models (1) to (3) tell a coherent and robust story about the geography of the platform labour market. The larger a region's population, the more projects it attracts. Similarly, higher levels of education are associated with higher project counts. The income level is negatively associated with project count on the global scale and in the OECD+, while it is positively associated in Global South regions. This indicates that it is middle income countries or regions, not the poorest places on earth, that attract most remote jobs. Internet connectivity is positively associated with project count: the better the internet infrastructure, the more remote platform work. The same holds the 'IT specialisation of the economy'. English language countries attract more projects, as well as those that have a lower price level. In Global South regions, the capital region indicator is strongly positively associated with activity.

The models (4) to (6) give insights into the factors that drive hourly wages. Overall, the coefficients show a similar direction as in the models (1) to (3), with a few important exceptions. For example, the English language coefficient is negative in model (4) because many high-wage countries in Europe do not have English as an official language. The 'IT specialisation of the economy' coefficient is negative in model (4). Income is positively associated on the country level and the price level is not significant (potentially because platform workers in countries with higher price levels tend to charge more). Education and internet connectivity are positively correlated with hourly wages.

The regression models explain a large share of the total variation (see $R^2$ values in Fig 2A and 2B). Almost three-quarters of the total variance between countries (left panel of Fig 2B) can be explained by the model: the residuals of very large players in the online labour market, such as India, the United States, or Ukraine (red dots) are almost zero. Similarly, model 4 accounts for 44% of global wage differentials (right panel of Fig 2B).

In summary, we conclude that the global distribution of remote work seems to be constrained by place-bound economic, infrastructure, and educational factors (RQ 2). The most profitable projects are conducted in places with high human capital levels, specialised know-how, and a robust local economy. Price differentials explain only a minor share of the global geography (S6 Appendix: S6.2 Section in S1 File).

## Polarisation across skills

To explain the wage differentials between job types in the platform labour market (Research Question 3), we look into the skills related to the professions. Following the task-based

approach [34, 37, 71], we consider jobs as the manifestation of different tasks and skills. Fig 3A displays the skill composition of each of the 46 occupations the platform job types can be grouped in (see S4 Appendix: S4.5 Section in S1 File for details on the matching between online job types and official occupational categories). Skills (rows) and occupations (columns) are sorted by a hierarchical clustering algorithm (see S4 Appendix: S4.5 Section in S1 File) to group occupations with similar skill requirements, resulting in nine skill- and six occupation cluster.

The upper half of the heatmap reveals significant differences between the occupations. Job types in clusters 3 and 4 score heavily on computer-related skills; those in clusters 2 and 6 score most intensively on language and (written) communication skills. Occupations in clusters 1 and 5 score most heavily on oral communication or clerical tasks.

The varying skill requirements translate into different hourly wages (lower part of Fig 3A). Jobs in clusters 3 and 4 pay an average wage of $ 16—17, jobs in cluster 2 and 6 an average wage of $ 15—16, and jobs in cluster 1 and 5 a mere average of $ 6—9 per hour. In other words, skill sets determine wages. This is confirmed by two regression models in Fig 3B). Model (1) relates the average hourly wages per occupation to three variables: the average number of applicants per project (competitive intensity), the total number of projects per occupation (size of each occupational sub-market), and the educational attainment score (reflecting differences in required education levels, see S4 Appendix: S4.5 Section in S1 File). The model identifies a strong negative association between wages and competitive intensity. The educational attainment score is positively associated with wages, particularly for 'Non-Tech' occupations (see panel (i) and (ii) in Fig 3C).

For platform workers, expected wages are essential but not the only relevant outcome. Due to uncertainty in the remote labour market, workers need to send quality signals to demonstrate experience and trustworthiness to potential employers, such as ratings or reviews about past projects [67, 72]. Platform workers that can secure some initial projects to distinguish themselves from other, less experienced, competitors might be able to obtain more profitable projects in the future. To operationalise the varying importance of experience signals in different occupations, we developed a statistical measure of the relevance of past experience in obtaining additional projects: the *experience gradient*. We calculate the experience gradient per occupation as the slope parameter estimate $\hat{\beta}$ of a regression between the number of projects a platform worker in occupation *i* obtained in year *t* by the number of projects the same worker had conducted in previous years (see S4 Appendix: S4.5 Section in S1 File). The experience gradient is related to occupation-level variables in model (2) of Fig 3B. The model shows that competitive intensity, measured by the market size, and average wages are positively associated to the relevance of experience. This applies more strongly to Non-Tech jobs (panels (iii) and (iv) in Fig 3C): in more competitive occupations with less skill-based signalling options, experience is more relevant than in others.

## Interpretation of the results

Here, we provide an interpretation of the quantitative findings regarding the spatial and skill-based polarisation or remote platform work. With respect to the geographical distribution (Research Questions 1 and 2), we argue that remote work does not simply flow to places with lower price levels, but that it is pulled towards locations with competitive advantages and a specialisation in information and communication technologies. This could be related to centripetal forces that characterise the spatial organisation of the institutions enabling remote knowledge work. These forces seem to be bigger than global price differentials, which could push remote work to places with low living costs. The places with the lowest

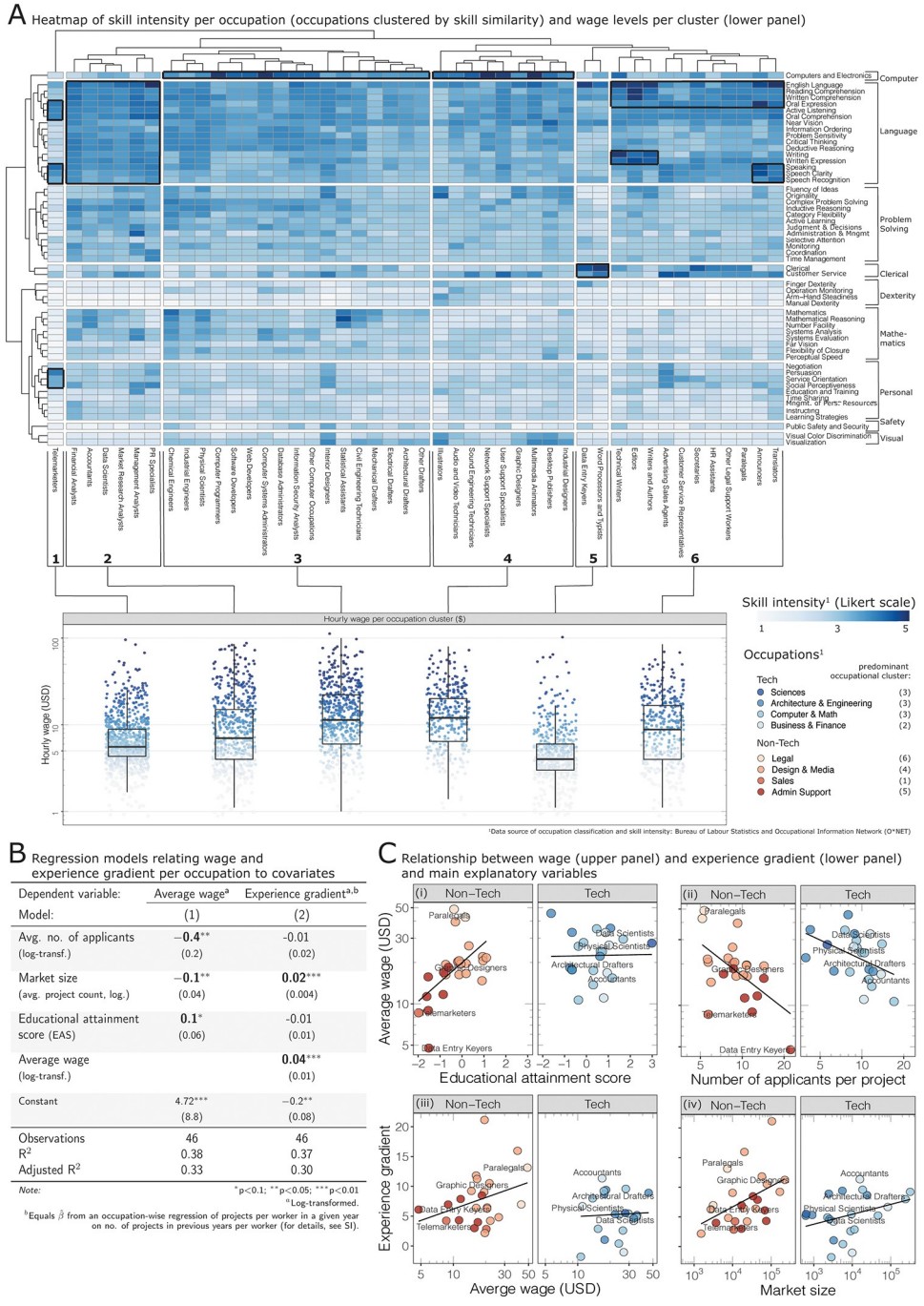

**Fig 3.** (**A**) Heatmap of skill intensity per occupation (upper panel) and wages per occupation cluster (lower panel): occupations (columns) with similar skill intensities (rows) cluster together; the highest paying occupations require computer-related know-how or English language comprehension and writing; lower-paying occupations focus on clerical skills, personal and oral communication. (**B**) Regression models between hourly wage (model 1), experience gradient (model 2) and occupation-level covariates: occupations with less competition (fewer applicants per job and lower avg. project count) and higher educational requirements pay higher average wages; previous experience counts more in high-paying occupations with fiercer competition (high project count). (**C**) Relations between wage (upper panel), experience gradient (lower panel) and occupation-level covariates: platform workers with only a fewer previous projects find it hard to be hired in non-tech occupations (panel iv).

price levels globally tend to not have a sufficient level of internet infrastructure, economic specialisation and know-how to enable workers to participate in the global market for remote knowledge work. The global polarisation in the remote labour market is then a digital mirror of the global polarisation of skills and economic opportunities across the globe.

Considering the results with regards to the distribution of skills and wages in the online labour market, we conclude the following. First, in contrast to conventional labour markets, which are shaped by geographical and regulatory constraints leading to substantial wage differentials for similar types of jobs even in close geographical proximity—think of wage differentials at the US-Mexican border region, Hong Kong vs Mainland China, Switzerland vs adjacent European countries or South Europe vs North Africa—the remote platform labour market is truly global. It is, however, not just one market, but many: one market per occupation. The variation in hourly wages between occupations is larger than the overall variation between countries (S6 Appendix: S6.4 Section in S1 File).

Secondly, the regression models show that jobs with more applicants and less skill-based signalling pay lower wages (Research Question 3). We interpret this finding as follows: These occupations have low entry barriers and face more competition. Without skills as a quality signal, the number of prior projects or feedback scores becomes an entry barrier. Employers use trust cues to decide whom to hire from the large crowd of applicants. For platform workers, this could spark a race to the bottom: without reputation, they find it hard to get their first job, so they need to undercut wages, leading to a vicious circle of more competition and lower wages.

In contrast, people with in-demand skills can secure profitable projects and high wages. The skill-based polarisation does not work along a one-dimensional skill axis: for example, the highest paying occupation is 'Paralegals and Legal Assistants'. This occupation does not require an exceptionally high level of formal education, but knowledge of the legal system in the country of the employer (in many cases the United States). Similarly, 'Announcers' (i. e. audio online adverts etc.) receive high wages. This occupation comes with another hard-to-copy skill: an U. S. accent. In general, jobs with hard-to-acquire, technical skills show less competition and higher wages.

Overall, the findings lead to the conclusion that demand for and supply of skills drive outcomes in the remote labour market. The three axes of polarisation—global divides between countries, urban-rural imbalances within countries, and inequalities between occupations—reflect the scarcity and abundance of skills and the access to them. In the global platform labour market, the laws of supply and demand work unrestrained: individuals with in-demand skills are able to secure profitable jobs; others obtain low wages, face fierce competition, and reputation as a crucial entry barrier. The outcomes of individual platform workers are constrained by system-level mechanisms shaped by agglomerative forces, which are largely out of their hands. The jobs remote workers can perform online is then determined by their access to education, training, and specialised IT know-how. This access is linked to place-bound institutions of the local economy. If they are unlucky not to be located near specialised industries or agglomerations, they are more likely to offer work in occupations characterised by easy-to-copy skills and fierce competition. In contrast, remote workers from metropolitan areas already have access to ample urban opportunities for knowledge exchange and local work opportunities, due to a bundling of complex economic activities [22] and a more fine-grained division of labour in urban environments [60]. They will enjoy the increased global demand for IT and business services and attractive wages.

## Discussion

The Covid-19 pandemic has led to the rise of remote work. Digital technologies enable the practical organisation of work at a distance. The potential cost savings, more flexibility, and improved access to talent suggest that remote work is likely to play an essential role in the future of work. However, it is unclear how far remote work will influence the global division of labour. In particular, does remote work bring jobs to rural areas and disadvantaged regions or will it reinforce global spatial imbalances? To investigate this question quantitatively, we draw on data from a fully remote labour market: an online labour platform. In the empirical part of this study, we analyse remote work mediated by an online labour platform. Here, in the discussion section, we also consider what implications the quantitative findings may have for remote work more generally, including regular employment performed remotely over the internet.

On the platform, workers from all over the world can find and conduct jobs covering the whole spectrum of knowledge work. As the whole work process—from the job advert over the interview, onboarding, communication, to payment and dispute resolution—is conducted online, the platform labour market could provide an outlook into the future of work. This future might be shaped by more remote contracts [73, 74] and the platformisation of jobs [75–77].

The data suggest that agglomerative forces shape the global geography of remote work, leading to the following interpretation. Jobs are pulled to places with enabling institutions of remote work, which are unequally distributed across the globe. Complex economic activities and specialised vocational training concentrate in large cities [22, 42]. In having access to these opportunities and skills, the remote workforce in urban areas is able to obtain the most profitable remote jobs, while their counterparts outside of economic centres find it more difficult to offer in-demand skills on the global platform labour market. The unequal spatial distribution of skills, institutions, and opportunities determine the global geography of remote work. The dynamics of the platform economy amplify this process, as there are little geographical or regulatory boundaries on the online platform that would slow down the global competition.

Across countries, we observe a spatial division of work that resembles the offshoring rationale of business processes, which started in the 1980s and 1990s [4]. Increasingly modularised and standardised tasks within the ever-growing digital economy have enabled a fine-grained global division of knowledge work connecting North America, West Europe, and Australia with South Asia, the Philippines, and East Europe. This observation is in accordance with earlier work that investigated the increasingly globalised market for knowledge work and Digital Taylorism—the modularisation and standardisation of cognitive tasks together with a fine-grained division of these tasks across different job types in global digital value chains—such as 'The Global Auction' by Brown et al. (2010). However, the data shows that most countries in the Global South are only marginally connected to the global web of remote work in the platform labour market.

Within countries, we find that remote work flows to urban centres. These are the places where highly skilled labour is concentrated. The economic tale of the 'booming metropolis' and the 'broken provincial city' [40] plays out fully in the platform economy. This imbalance resembles existing opportunity gaps between urban and rural areas, which have pulled skilled workers to cities already before the rise of the platform economy. In fact, while online labour platforms are thought to offer an alternative to employment-based migration, constituting a form of 'virtual migration' [78], studies suggest that online labour platforms are frequently used by migrants [79]. Similarly, online labour platforms are more intensively used by younger parts of the working age population. Pajarinen et al. (2018) find that many online freelancers in Finland are less than 30 years old [79]. It is the young, talented part of the workforce that

most likely migrates to urban areas because of better income opportunities [80]. Such opportunity-based migration to metropolitan areas and resulting differences in the age structure of urban and rural areas is probably one of the drivers of the geographical disparities in the platform data. Overall, we observe that remote platform workers in metropolitan areas are more likely to attract specialised online jobs and high wages.

The findings highlight the pivotal role of skills in driving remote work towards metropolitan areas. Individual occupations form sub-markets of the global platform labour market. Platform workers are constrained to work in those jobs that reflect their skills and experiences [81]. Competitive pressure differs between job types as workers cannot freely move between occupations. This leads to a scarcity premium in some occupations, while others suffer from low wages and excess supply. In that situation, the uncertainties of the platform economy spark a race to the bottom: feedback and prior work experience are highly relevant to obtain remote jobs [46, 82]; newcomers will find it hard to get their first job and might be forced to undercut wages. This fuels competition and spirals wages downwards. Besides the intense competition and limited opportunities for signalling work quality, which we describe in this study quantitatively, the literature has discussed the role of algorithmic control [27, 83] and the organisation mechanisms of work in the platform economy [30, 84] as drivers of (adverse) outcomes for remote platform workers.

Our analysis implies that agglomerative forces drive the polarisation of the remote labour market. Market access alone will not lead to a less unequal division of labour. Remote workers in disadvantaged areas need more than a computer and broadband internet alone to thrive. They need to have marketable skills to make remote work a tool for rural and economic development.

## Policy Implications

Initiatives such as the Rockefeller Foundation's *Digital Jobs Africa* or Kenya's *Ajira Digital* work programme aim to bring millions of remote jobs to Africa, but they could make matters worse for remote workers: if they increase the supply in certain types of occupations, they could fuel the competitive spiral of excess labour supply and pressure on wages. To increase chances of remote workers in disadvantaged regions, retraining programmes need to focus on in-demand skills and account for the quickly changing dynamics in the global market for talent. It is unlikely that remote worker communities can thrive if there are limited local opportunities. Therefore, online work programmes in rural areas—both in Global North and Global South countries—should be embedded in larger economic and labour market development schemes, which provide reliable internet access, local employment alternatives, and skill-building opportunities. This applies also to remote labour demand. Remote platform work can be a chance for rural employers to get access to global talent. Programmes that foster the integration of remote work into their business processes might help companies to become more resilient and to keep them in their local surrounding.

Online platform providers could increase the visibility of objective quality metrics, such as educational degrees, to limit the adverse effects of reputational feedback loops in the remote platform labour market. Platform apprenticeships for new remote workers—the random assignment of first jobs to people without experience on the platform—could help to build up initial credibility [67] and lower entry barriers. Moreover, governmental organisations that aim to improve the working conditions of remote platform workers, such as the European Commission, could support the positive development of platform work in advertising short-term remote jobs directly on online platforms while promoting living wages. They could also help in developing and accrediting objective quality metrics. Another way to strengthen local

communities of remote platform workers could be the support of coworking spaces and other forms of physical meeting points for platform workers. Such spaces could focus on providing workplaces with all the equipment and services necessary for performing remote jobs effectively. In bringing remote workers together and offering complementary services, these spaces could foster knowledge exchange and skill-building, and they could help remote platform workers to gain a sense of community.

### Implications for the future of remote work

Why is it that remote work is unlikely to change the economic imbalance between urban and rural areas majorly? Remote work allows people to move freely from urban to rural areas only in the short term. Some urban specialists might enjoy the new remote work opportunities and relocate to suburban or rural environments, substituting day-to-day commutes with digital interactions. They can stay connected to their peers in urban centres via video calls, but they will find it difficult to establish new links and access informal knowledge exchange through local networks.

In contrast, the city, as a hub of interaction supports specialised local jobs and occupational diversity [60, 85, 86]. An increasing share of complex economic activities [22], and non-linear scaling laws make the provision of specialised occupations more sustainable in large agglomerations [87–89]. This will likely continue to pull business opportunities and people towards cities. Digital technologies and organisational adjustments during the Covid pandemic have enabled seamless communication and collaboration over distance, which theoretically allows for a wide-spread web of economic and labour market interactions between urban and rural environments. Still, the network forces that pull innovation and business opportunities towards large agglomerations are likely to also shape the global geography of remote work. In the remote labour market, the place of work might not be limited to the same city anymore. But instead of dispersing more equally across space, remote work probably tends to cluster in metropolitan areas in different parts of the world, which share similar institutions and urban lifestyle that enable knowledge work to flourish locally.

### Limitations

Our study comes with some methodological limitations. The data collection (described in S4 Appendix in S1 File) dependents on access to the online platform's API. We cannot make claims about the size of our sample in relation to the overall size of the remote platform labour market, but we are confident that sampling issues did not bias the analysis. This is because our findings are robust over all the years in the sample and they align with previous investigations on the geography of platform work. Furthermore, Our study analyses data from only one platform, but the platform investigated here is one of the global market leaders. In the data analysis, we had to make simplifying assumptions: in mapping the platform job categories to the official occupation taxonomy, we had to disregard the multifaceted skill-dimensionality of jobs within each occupation. Moreover, the algorithmic geocoding and occupation mapping come with some uncertainties. However, we very carefully investigated each step of the data preparation for potential errors (outlined in the SI) and we performed several robustness checks (for example, S6 Appendix: S6.1 Section in S1 File) to validate parameter choices.

### Conclusion

Remote work mirrors the spatial inequalities of labour markets at large. The most profitable jobs are pulled towards the booming tech-savvy metropolis, while rural areas fall behind. In contrast to on-site labour markets, the polarisation mechanisms are amplified in the platform

economy, as the forces of supply and demand are fully unleashed in the absence of regulatory barriers.

The unequal spatial distribution of institutions enabling skill-building and business opportunities determine the geography of the remote labour market. While internet connectivity and price differentials channel remote work around the globe, only remote workers with access to specialised skills attract valuable projects. Platform reputation mechanisms further accelerate the global race to the bottom for those who do not possess in-demand skills. Still, remote work can become an instrument of economic empowerment and growth. For this to happen, remote work needs to be embedded in broader economic and labour market development schemes, supporting disadvantaged regions to invest in local skill development and infrastructure.

Only in regions that flourish locally, remote workers can succeed globally.

## Supporting information

**S1 File.**
(PDF)

**S1 Text.**
(PDF)

## Author Contributions

**Conceptualization:** Fabian Braesemann, Mark Graham.

**Data curation:** Fabian Braesemann, Otto Kässi.

**Formal analysis:** Fabian Braesemann, Fabian Stephany, Ole Teutloff.

**Funding acquisition:** Fabian Braesemann, Mark Graham, Vili Lehdonvirta.

**Investigation:** Fabian Braesemann, Fabian Stephany, Ole Teutloff, Vili Lehdonvirta.

**Methodology:** Fabian Braesemann, Fabian Stephany, Otto Kässi, Vili Lehdonvirta.

**Project administration:** Fabian Braesemann.

**Resources:** Fabian Braesemann, Otto Kässi, Mark Graham.

**Software:** Fabian Braesemann, Fabian Stephany.

**Validation:** Fabian Braesemann, Fabian Stephany, Ole Teutloff, Otto Kässi, Mark Graham, Vili Lehdonvirta.

**Visualization:** Fabian Braesemann, Fabian Stephany.

**Writing – original draft:** Fabian Braesemann, Fabian Stephany, Ole Teutloff, Otto Kässi, Mark Graham.

**Writing – review & editing:** Fabian Braesemann, Fabian Stephany, Ole Teutloff.

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
