## [Decision Letter · Decision Letter 0]

11 Mar 2022

PONE-D-22-02805The global geography of remote workPLOS ONE

Dear Dr. Braesemann,

Thank you for submitting your manuscript to PLOS ONE. After careful consideration, we feel that it has merit but does not fully meet PLOS ONE’s publication criteria as it currently stands. Therefore, we invite you to submit a revised version of the manuscript that addresses the points raised during the review process.

We look forward to receiving your revised manuscript.

Kind regards,

Hocine Cherifi

Academic Editor

PLOS ONE

Journal Requirements:

Reviewers' comments:

Reviewer's Responses to Questions

**Comments to the Author**

1. Is the manuscript technically sound, and do the data support the conclusions?

Reviewer #1: Yes

Reviewer #2: Yes

2. Has the statistical analysis been performed appropriately and rigorously? 

Reviewer #1: Yes

Reviewer #2: Yes

3. Have the authors made all data underlying the findings in their manuscript fully available?

Reviewer #1: Yes

Reviewer #2: Yes

4. Is the manuscript presented in an intelligible fashion and written in standard English?

Reviewer #1: Yes

Reviewer #2: Yes

5. Review Comments to the Author

Reviewer #1: Well-written, clearly structured and, obviously, a very timely contribution to the debates on the changes in the geography of work. Many thanks for that. Just a few minor suggestions.

1) The positioning of the case (the global labour platform) now only occurs in the discussion on the limitations, it should be more prominent: is this representative of other platforms? Of all online work arrangements?

2) The findings in terms of correlations between the selected variables seem to me very robust. There is, however, one potentially, important variable missing: age. It might be that the strength of the agglomeration economies is much more significant for those at the start or their career, whereas those who have been able to carve out a niche might already have transcended the don't call us we'll call you phase

3) There is a reference to the centre-periphery pattern as if this has been constant in the past the decades. The emergence of India and the Philippines are proof of fundamental shifts in this.

4) The segmented market for online workers, resembles quite closely the dual labour that we once had in developed, Fordist economies.

5) What about worker spaces as physical meeting points for online workers?

Reviewer #2: This is an interesting paper offering new insights on the platform labour market using a methodological approach. The paper is let down by the theoretical framing and the discussion of the findings

There is also a fundamental confusion about remote work and platforming work. Many white collar employees may be able to work remotely (at least some of the time). This does not mean that they are likely to organise work through online work platforms. The paper deals with work organised via online platforms (and is therefore remote) not remote work in general. In various point is the paper it equates the two and that is not justified. For instance, it starts off by stating ‘that remote working is the new normal’. Apart from the fact that this is an exaggeration and it can be much more precise in telling how widespread remote working is, the paper seems to frame the topic of the paper as one about remote work. It then state that ‘Remote work organised via online platforms could bring jobs to workers from all over the globe’. This is true in theory but in practice, most work remote or not is not going to allocate through online platforms. Similarly when the authors state that “the whole spectrum of knowledge work” can be performed remotely, that is arguably not true realistically. It also depends on that depends on how you define knowledge work. Rather than remaining rather overstating the reach of platfrom labour markets It would strengthen the paper by being much more clear what type of jobs and skills are likely to be performed through the platform labour market. The introduction rightly explains that the growth of platform labour market will affect the earnings and opportunities work workers in both global north and south. Here again it can do much more to contextualise this. It mentions digital Taylorism which I think is a good way to understand some of the effect of so-called knowledge work. It mentions Brown et al and their conceptualisation of the digital taylorisims but does not offer a detailed enough account of their argument. It writes about the

“global imbalance between the excess supply of highly educated graduates in Global South countries and the high demand for talent in the Global North. “

Yet according to Brown et al there is also an excess supply of graduates in the North and the war for Talent is globally organised and not limited to the North.

The introduction can also be much clearer in the language it is using. For instance it mentions

“geographical frictions and biases that restrict participation” and “Similar to other complex economic activities remote platform work might cluster in large cities” Explain to the reader what these ‘frictions and biases’, and ‘complex economic activities’ are or give some examples

Similarly when it states that, “in the absence

of sufficiently granular data, our understanding of the global geographies of platform-mediated

remote work remains limited.”, the reader would like to what is it that we need to understand and why do we need granular data.

Explain what you mean by “agglomerative forces” ?

The introduction also needs be more clear what are findings and what are interpretations. When it states that “The antagonism

85 between the ’booming metropolis’ and the ’broken provincial city’ [39] plays out fully in the remote

86 labour market, as the institutions that enable a successful participation—access to knowledge

87 building, training and professional networks—concentrate in urban environments. Rural regions

88 are not able to offer specialised work opportunities and urban lifestyle [40, 41].” That is not grounded in the data and rather speculative. This is not to say these are credible explanations but it can needs to be clear to the reader

“Under these conditions, market outcomes are driven by imperfect information, uncertainties, trust

92cues, and reputation systems” This needs needs to be explained or left out. All labour markets by (some of) these.

We argue that the unequal global distribution of remote work is the result of the unbalanced

98 distribution of skills, human capital, and opportunities across the globe [47, 48]. This uneven dis99

tribution of economic conditions and competitive advantages transcends to the platform economy

100 and drives the geographical polarisation of the remote labour market.

The literature review is quite descriptive and offers an overview of offshoring but not an analysis of the literature, either theoretical or empirical research research on the role of ICT. How does the

The review is rather short and a more in-depth and expansive discussion on the literature on global polarisation of remote platform work would make much clearer later on how the study extent the current literature.

Also again, a lack of clarity in the writing

On line 130 it states that growing evidence that utilising the power of digital technologies requires complementary skills and capabilities

What complementary skills and capabilities?

It could do better in supporting the portrayal that urban centres equate with vibrant business eco systems? Can the authors give evidence for this? It is believable that skills concentrates are in/near urban centres, but not all urban centres are necessarily skilled ecosystems. How it is written seems overly impressionistic rather than a robust overview of existing literature

The paper states later on that “online labour platforms represent one avenue for the integration of remote workers into global

141 digital value chains through outsourcing and offshoring. As such, they showcase how digital

142 technologies can reshape economic geographies”

Again, they can, but how, and to what extent are they reshaping economic geographies.

A set of research questions at the end of the literature review would help the reader

Methods

The paper draws on high-quality data. The methodology seems suitable for the aims of the study

Minor: “The data collection methodology is an essential part of this study” Odd sentence

Findings

The findings are interesting and important. Here it could be more explicit. For instance

“The global polarisation resembles core-periphery structures well-known from other

domains of the platform economy”

It is not clear what they are and why

. Within the findings there is the kind of interpreatation one normally finds in the discussion Also, be careful not to confuse causation with the association. The language is often interpretative (contrain/enabling)For instance,

“The global polarisation in the remote labour market is the digital mirror

324 of the global polarisation of skills and economic opportunities across the globe.” This is strictly not a finding, or

“The jobs remote workers can

390 perform online is determined by their access to education, training, and specialised IT know-how.

391 This access is linked to place-bound institutions of the local economy. If they are unlucky not

392 to be located near specialised industries or agglomerations, they will be more likely to offer work

393 in occupations characterised by easy-to-copy skills and fierce competition. In contrast, remote

394 workers from metropolitan areas already have access to ample urban opportunities for knowledge

395 exchange and local work opportunities.”- this is rather speculative and should not be part of the findings (but could be part of the discussion, if it clearer which assumptions are being made) , where those skills have been developed

Discussion

In the discussion remote work is equated with platform work. It needs to make a distinguish the two It also states that

the platform labour market provides an outlook into the future of work, in

408 which fully remote contracts and platformisation might be the norm.

This needs to be discussed better

No engagement with literature on capitalism or digital Taylorism which is as shame. It also does not discuss the role of migration and how it affecta the finding. Skilled or motivated workers may move for various reasons to cities as a result more skilled workers are to be found in urban areas perhaps partly driven by the non-remote economy and its labour market opportunities.

6. PLOS authors have the option to publish the peer review history of their article (what does this mean?). If published, this will include your full peer review and any attached files.

Reviewer #1: **Yes: **Robert C. Kloosterman

Reviewer #2: No

---

## [Author Response · Author response to Decision Letter 0]

7 Jun 2022

Dear Editor, 

Thank you for giving us the opportunity to submit a revised version of our manuscript "The global geography of remote work" and thanks to the reviewers for all the useful comments being made by them. Please find our detailed point-by-point responses to the comments in the cover letter of this submission. After the cover letter, the submission contains the original (revised) manuscript without track changes, then a link to the supplementary information and then the revised manuscript with track changes (highlighted in green).

---

## [Decision Letter · Decision Letter 1]

1 Sep 2022

The global geography of remote work

PONE-D-22-02805R1

Dear Dr. Braesemann,

We’re pleased to inform you that your manuscript has been judged scientifically suitable for publication and will be formally accepted for publication once it meets all outstanding technical requirements.

Kind regards,

Hocine Cherifi

Academic Editor

PLOS ONE

Additional Editor Comments (optional):

Reviewers' comments:

Reviewer's Responses to Questions

**Comments to the Author**

1. If the authors have adequately addressed your comments raised in a previous round of review and you feel that this manuscript is now acceptable for publication, you may indicate that here to bypass the “Comments to the Author” section, enter your conflict of interest statement in the “Confidential to Editor” section, and submit your "Accept" recommendation.

Reviewer #1: All comments have been addressed

2. Is the manuscript technically sound, and do the data support the conclusions?

Reviewer #1: Yes

3. Has the statistical analysis been performed appropriately and rigorously? 

Reviewer #1: Yes

4. Have the authors made all data underlying the findings in their manuscript fully available?

Reviewer #1: Yes

5. Is the manuscript presented in an intelligible fashion and written in standard English?

Reviewer #1: Yes

6. Review Comments to the Author

Reviewer #1: This is an important contribution to the debate on digital labour markets. The study provides an in-depth analysis of the expanding Global though highly spatially skewed digital labour market. It clearly shows how on line work demonstrates a distinct geography - on the level of global regions, countries and within national states. it should be seen as a crucial step to further research which should explore the role of agglomeration economies in concentrating digital workers in certain metropolitan areas. The authors do mention a few of them, but - given the age profile of these workers, most of them young adults - you would expect that their preference for cities might also have to do with a large pool of potential partners who are also highly educated and more internationally oriented. This should be, however, a next step in the investigations regarding remote work. Another issue that might be covered in that follow-up research is the question to what extent the metropolitan location is permanent - both on a short-term basis as people may move temporarily to other places (say in the mountains or the countryside) and, later on, when people get older and have established a relatively secure client base and are able to move out to greener (and cheaper) pastures.

7. PLOS authors have the option to publish the peer review history of their article (what does this mean?). If published, this will include your full peer review and any attached files.

Reviewer #1: **Yes: **Robert C. Kloosterman

---

## [Editor Report · Acceptance letter]

11 Oct 2022

PONE-D-22-02805R1 

The global polarisation of remote work 

Dear Dr. Braesemann:

I'm pleased to inform you that your manuscript has been deemed suitable for publication in PLOS ONE. Congratulations! Your manuscript is now with our production department. 

Kind regards, 

on behalf of

Professor Hocine Cherifi 

Academic Editor

PLOS ONE